# Deciphering Exhaled Aerosol Fingerprints for Early Diagnosis and Personalized Therapeutics of Obstructive Respiratory Diseases in Small Airways

Xiuhua April Si [1] and Jinxiang Xi [2],*

1   Department of Aerospace, Industrial and Mechanical Engineering, California Baptist University, Riverside, CA 92504, USA; asi@calbaptist.edu
2   Department of Biomedical Engineering, University of Massachusetts, Lowell, MA 01854, USA
*   Correspondence: jinxiang_xi@uml.edu; Tel.: +1-989-774-2456

**Abstract:** Respiratory diseases often show no apparent symptoms at their early stages and are usually diagnosed when permanent damages have been made to the lungs. A major site of lung pathogenesis is the small airways, which make it highly challenging to detect using current techniques due to the diseases' location (inaccessibility to biopsy) and size (below normal CT/MRI resolution). In this review, we present a new method for lung disease detection and treatment in small airways based on exhaled aerosols, whose patterns are uniquely related to the health of the lungs. Proof-of-concept studies are first presented in idealized lung geometries. We subsequently describe the recent developments in feature extraction and classification of the exhaled aerosol images to establish the relationship between the images and the underlying airway remodeling. Different feature extraction algorithms (aerosol density, fractal dimension, principal mode analysis, and dynamic mode decomposition) and machine learning approaches (support vector machine, random forest, and convolutional neural network) are elaborated upon. Finally, future studies and frequent questions related to clinical applications of the proposed aerosol breath testing are discussed from the authors' perspective. The proposed breath testing has clinical advantages over conventional approaches, such as easy-to-perform, non-invasive, providing real-time feedback, and is promising in detecting symptomless lung diseases at early stages.

**Keywords:** exhaled aerosol fingerprint; lung diagnosis; personalized therapeutics; obstructive respiratory disease; nanoparticles; fractal; random forest

## 1. Introduction

Exposure to environmental and occupational toxins can lead to various respiratory disorders [1,2], such as pneumoconiosis [3], chronic obstructive pulmonary diseases (COPD) [4], and lung cancer [5]. Small airways in deep lungs can be significantly involved in the early course of the pathogenesis before the onset of symptoms [6,7]. However, remodeling in such small airways has proven difficult to be detected by existing diagnostic methods due to their distal location (inaccessibility to biopsy) and tiny size (below normal CT/MRI resolution). Considering the possible decade-long symptomless development of lung diseases, it is crucial to detect the subtle, silent pathogenesis at its early stages [8–11]. It is also highly desirable to detect the level of severity of the disease, as well as the site of the disease, so that targeted intervention can be formulated.

Early diagnosis of lung cancer is crucial to patient survivability. The cure rate can be 70% for patients with non-small cell lung cancer (NSCLC) if detected at stage I but only 25% if detected at stage III [12,13]. Current techniques to detect lung cancer and respiratory diseases include spirometer for pulmonary function tests, X-ray for screening, SPET/PET/CT to identify airway remodeling, and biopsy to evaluate the disease type and extent [14]. These diagnostic tools are reliable in general but are also expensive and

require professional operations. Some have radiological risks (CT/PET/SPET) and can be invasive (biopsy).

Pneumoconiosis is the most common respiratory disease among coal workers [15]. Prolonged exposure to airborne dust, particularly mineral dust, can progressively lead to airway inflammation, fibrosis, black lung, or even death [16]. However, patients with pneumoconiosis may have no apparent symptoms at their early stages and are usually diagnosed by routine workplace surveillance tests when symptoms are obvious and significant damages have been made to the lungs [17]. Because there is no cure for pneumoconiosis, early detection of this disease is critical to the patients' quality of life, as well as survivability. Patients with pneumoconiosis experience frequent episodes of shortness of breath, cough, and intolerance to heavy exercise [18], while patients with severe pneumoconiosis may have to live with external oxygen supplies or undergo lung transplants. In addition, patients with pneumoconiosis and silicosis are at a higher risk of developing tuberculosis and rheumatoid arthritis [19].

In recent years, new diagnostic devices for lung diseases have surfaced that utilize exhaled breath, which contains clues of lung health [20–23]. A growing tumor cell is also accompanied by metabolic changes, which produce a unique group of chemicals and form a signature breath "fingerprint" for that type of tumor. This breath fingerprint can be exploited to decide if a specific pathology is present and if so, what stage it is in [24]. As the breath test provides a non-invasive alternative to conventional diagnostic tools, extensive studies have been carried out in analyzing the breath contents and identifying signature breath biomarkers. Increased concentrations of nitric oxide have been associated with asthma [25], cytokines with cystic fibrosis [26], antioxidants with chronic obstructive pulmonary disease (COPD) [27], while decane, $H_2O_2$, and isoprene with non-small cell lung cancer (NSCLC) [28,29]. While some biomarkers have already been applied in clinical practices, such as nitric oxide for asthmatic patients, other biomarkers are still being developed (e.g., profiles of volatile organic compounds—VOC) [30]. Reviews of using VOC breath tests for lung cancer diagnosis and developments of associated medical devices are presented in [23,31–33]. One critical limitation of the gas-signature-based approach is that it can only determine the presence and concentration of certain exhaled chemicals. No information on the carcinogenic site can be retrieved from such breath tests, which is also essential to cancer treatment planning. At the current stage, such information can only be attained using imaging tools such as MRI, PET, or CT, which, despite high accuracy, suffer from high cost and potential health risks [34,35]. Moreover, targeted delivery could only live up to its name if the diseased tissue was found beforehand. It is, therefore, highly desirable to have alternative tools that can pinpoint the malignant sites in a non-invasive and less costly manner.

Computational simulations of respiratory dynamics have persistently indicated that the distributions of exhaled airflow and aerosols, although appearing chaotic in pattern, are indeed unique to lung physiology [36–38]. Whenever there is a structural remodeling in the lung, these distributions change accordingly. It is therefore hypothesized that each lung generates its signature exhaled aerosol fingerprint (AFP). The AFP pattern is a collective distribution of all exhaled trace particles that have traveled through the respiratory tract. As such, a deviation from the normal profile indicates airway remodeling within the respiratory tract and it should be possible to retrieve it using the inverse approaches that were proposed in Xi et al. [37–43].

The proposed concept of AFP-based breath testing has two characteristics that are desirable in lung disease diagnosis and treatment: (1) detect and locate the disease site, and (2) deliver a personalized dose of therapeutic agents to the diseased site. If the concept were proven feasible, an integrated diagnostic–therapeutic system could be further developed. The device would be non-invasive, give real-time diagnosis feedback, be capable of precise drug delivery, and at the same time, it would be easy to use and low-cost. The vision is that the patient takes a breath test for diagnostic purpose, and a patient-specific drug delivery protocol can be subsequently developed, targeting therapeutic aerosols at the diseased

site only, thereby optimizing the medical outcome and minimizing side effects. The proposed AFP-based breath testing will work best for respiratory diseases with measurable structural remodeling, such as asthma, COPD, and NSCLC. The proposed breath testing will be especially suitable for respiratory disease screening for those under elevated and prolonged exposures, such as workers in the textile industry, coal mine, wood industry, and refinery. Due to its easy-to-use and low-cost features, the workers could do the tests more frequently. Regular testing can track the changes in the patient's lung health. The ability to localize the disease sites can facilitate a personalized treatment protocol, such as targeted topical inhalation therapy.

Previous studies also attempted to use exhaled aerosols as diagnostic tools, based on a method termed aerosol bolus dispersion (ABD) [44–48]. The similarities and differences between the method presented in this study and ABD are described as follows. Both methods were based on the same concept—that a patient with a remodeled lung exhales differently, which manifests itself in expiratory fine-regime aerosols. However, at least three factors establish the novelty of the proposed method: (1) test procedures, (2) variables to measure, and (3) information that we can get from the test. Concerning the test procedures, ABD needs only one step, i.e., inhalation–exhalation. By contrast, the AFP-based test needs up to three steps. In the first step, a bolus aerosol is inhaled to a predefined depth and then exhaled for screening purposes. In the second step, aerosol tracers are released selectively to the suspected regions of the lung to localize the disease site. In the third step, a personalized treatment plan can be developed that targets pharmaceuticals at the malignant tissues only. Regarding the variables to measure, ABD records the aerosol concentration as a function of the flow volume to evaluate the severity of airway remodeling [44,47]; on the other hand, the AFP-based approach directly uses the images (particle distribution pattern) of the exhaled aerosols collected on a mouth filter. Lastly, concerning new information that can be inferred, ABD tests will not generate new information on lung health other than current functional tests. By contrast, new information on the disease site can be discovered from the proposed AFP-based tests, which is crucial in achieving targeted drug delivery to the diseased site.

In this study, we review the recent developments in feature extraction and classification of the exhaled aerosol images to establish the relationship between the images and the underlying airway remodeling. Different feature extraction algorithms (aerosol density, fractal dimension, principal mode analysis, and dynamic mode decomposition) and machine learning approaches (SVM, random forest, and convolutional neural network) are discussed, with emphasis on their suitability and accuracy in categorizing respiratory diseases based on exhaled aerosol images. In the end, potential roadblocks to clinical applications of the proposed aerosol breath testing for lung diagnostics and therapy will be discussed from the authors' perspective.

## 2. Materials and Methods

### 2.1. Healthy and Diseased Airway Models

In this study, images of exhaled aerosols, or AFP, were numerically generated using physiology-based modeling and simulations in anatomically accurate airway models, as shown in Figure 1a. To this aim, two respiratory airway models were developed. The first model was a simplified geometry that included the respiratory tract from the oral cavity to the sixth generation (G6) of lung bifurcations (Figure 1b,c), while the second model was a more complex, realistic geometry that extended to the ninth generation (G9) of lung bifurcations (Figure 1d). The first model was developed based on MRI scans of a 53 year old non-smoking male with no respiratory disease [36]. Mimics (Materialise, Ann Arbor, MI, USA) were used to segment the airspace from other organs. Due to the presence of artifacts in the segmented geometry, the polylines that enclosed it were extracted, and they were further used as the scaffold to reconstruct the airway surface geometry by patching it with ~3–5-edge faces. This method allowed cleansing of apparent artifacts, improved computational quality, and controlled modification of regional airway structures.

The downside of this method was that it was time-consuming and labor extensive, as most of the procedures, such as artifact-cleansing and surface-patching, were manual. There were 23 outlets in the geometry. Detailed procedures of model development and airway dimensions can be found in Xi and Longest [36]. The second model shared the mouth–throat geometry with the first one, which was connected to a more complex lung model [49]. The volume of the lung model was scaled to be 3.5 L, which is consistent with the functional residual capacity (FRC) [50,51]. There were 115 outlets in this lung model. The two airway models were further modified using HyperMorph (Troy, MI, USA) to generate airway geometries that represented different types of diseases with varying severities (Figure 1b–d). A high-quality computational mesh was generated for each airway model. Three test cases were designed to evaluate the performance of proposed algorithms, as described below.

In the first test case (Figure 1b), we aimed to test if there were significant differences in exhaled aerosol distributions in the upper airway between health (Model A) and diseased lungs (Model B–D). Specifically, Model B had a 10 mm sized tumor at the carina ridge of the trachea, while Model C had a 4 mm sized tumor located at G3 (i.e., segmental bronchus) in the left lower lobe. In both models, the ratio of the tumor sizes to the host airway diameters followed those in Segal et al. [52], who investigated the influences of the location and size of tumors on respiratory airflows. In Model D, there were two constricted bronchi in the left upper lobe (number 3 and 4 in Figure 1b) intended to represent local asthmatic airways.

In the second test case, we aimed to characterize the differences of the exhaled aerosol images due to a growing tumor at the segmental bronchus (G3) (Figure 1c). A squamous tumor results from uncontrolled growth of round cells [53]. They can grow in size and constitute cavities in the lung parenchyma. These kinds of tumors are often observed in central larger airways, either in the main bronchi or the major lobes [53]. In this case (Figure 1c), the tumor was at the segmental bronchus of the lower left lobe and had five different diameters to model the varying stages of the tumor [54].

In the third test case, we aimed to test whether the proposed diagnostic method could detect the geometrical changes in small airway with diameters less than 2 mm (Figure 1d). The rationale behind this test is that a diagnostic tool should be sensitive (and robust) enough to work reliably in clinical settings, which can have more compounding factors. To this aim, four levels of airway constrictions (A1, A2, A3, A4) were generated by progressively narrowing the diameters of the bronchioles at G7 in the lower left lobe using HyperMorph [55,56]. The minimal diameter and cross-sectional area of the constricted bronchioles, along with the bronchiolar volumes that are affected by the constriction, can be found in [56].

### 2.2. Acquisition of Exhaled Aerosol Images

Physiology-based simulations were undertaken to mimic the breath tests and generate the images of exhaled aerosol distributions. A bolus of aerosol tracers was inhaled by the patient to a certain depth and was subsequently exhaled to an aerosol collector positioned at the mouth opening. The aerosol bolus was generated using a stochastic scheme and consisted of 30,000 particles [41]. The respiratory airflow was simulated using the low Reynolds number (LRN) $k$-$\omega$ model [58], which has been demonstrated to capture flow transitions accurately [59].

A discrete-phase Lagrangian tracking model with near-wall treatment was used to track the particle trajectories [60]. The governing equation for the particle motion was expressed as in [36]:

$$\frac{dv_i}{dt} = \frac{f}{\tau_p C_c}(u_i - v_i) + g_i(1 - \alpha) + f_{i,Brownian} + f_{i,Lift} \tag{1}$$

where $u_i$ is the flow velocity, $v_i$ is the particle velocity, $f$ is the drag factor [61], $\tau_p$ (= $\rho_p\, d_p^2/18\mu$) is the particle transient time, with $d_p$ being the particle diameter and $\mu$ the flow viscosity, and $C_c$ is the Cunningham slip factor as formulated in [62]. User-defined functions (UDFs)

were applied to take into account the finite particle inertial [63] and near-wall damping [64]. This model had been well validated in our previous studies, which had provided good agreements with comparable in vitro measurements for both nano particles [63] and micron particles [38]. The computational meshes were created using ANSYS ICEM CFD (Ansys, Inc), grid independent study being performed with incrementally increasing mesh densities [37] until the variation in the variable of interest was less than 1%.

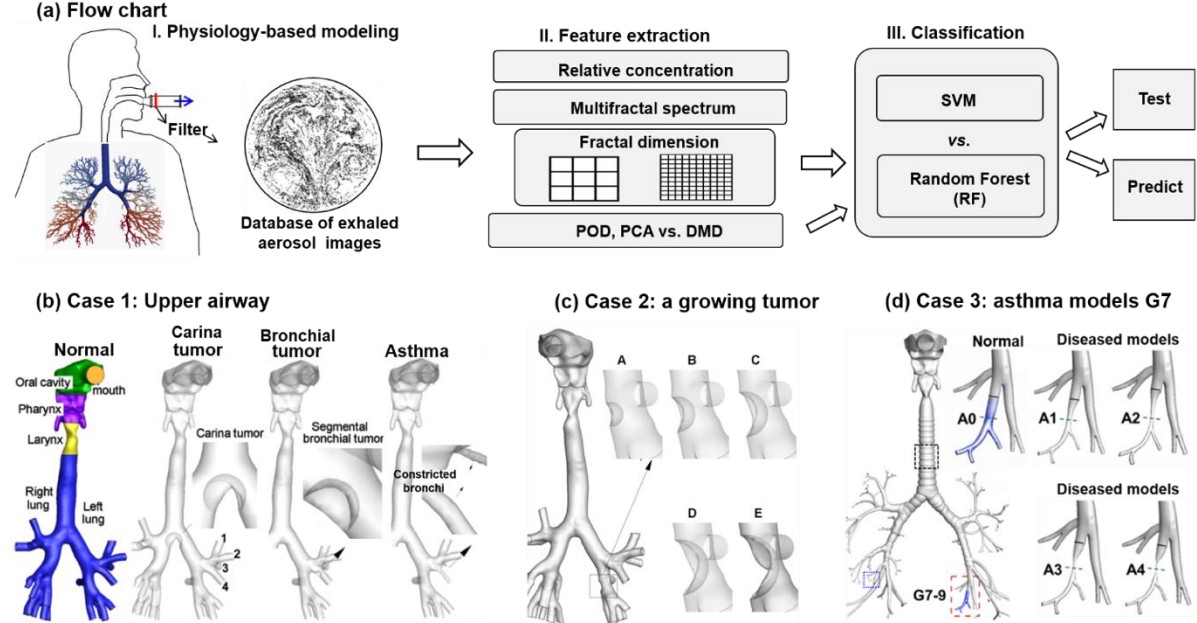

**Figure 1.** Study methodology and airway models: (**a**) flow chart of this study that consists of three steps: physiology-based modeling and simulation, feature extraction, and classification; (**b**) test case 1: normal vs. diseased models in large airways (adapted from [57]), (**c**) test case 2: a growing squamous tumor at the segmental bronchus (G3) (adapted from [54]); and (**d**) asthmatic model in small airways (G7) (adapted from [56]).

### 2.3. Feature Extraction of Exhaled Aerosol Images

#### 2.3.1. Relative Concentration

Although the particle distributions can be visually differentiated between different lung geometries, particle overlapping may make such visual judgment unreliable. To accurately quantify particle concentrations, the ratio of local aerosol concentration to the mean can be calculated [65]. For lung models with small structural variations, the exhaled aerosol distributions can be very similar. The quantitative aerosol concentrations can help identify such structural variations, which are also the sites of pathogenesis.

#### 2.3.2. Fractal Dimension and Multifractal Spectrum Analysis

The exhaled aerosol images exhibit a complex profile and can be characterized using the box-counting fractal dimension (DB), which quantifies the complexity of the image at varying resolutions. Mathematically, DB is computed as the gradient between the number of grids containing pixels (box-count) $N\varepsilon$ and the resolution scale $\varepsilon$ in the log–log plot.

$$D_B = \ln N_\varepsilon / \ln \varepsilon \qquad (2)$$

The multifractal spectrum analysis is another approach to measure the complexity of a system. This method rests on the fact that a natural system usually has multiple scaling probabilities. To compute the multifractal spectrum, a normalized parameter $\mu_i(q,\varepsilon)$ can be calculated from a group of scaling exponents, $q$ [66].

$$\mu_i(q,\varepsilon) = [P_i(\varepsilon)]^q / \sum [P_i(\varepsilon)]^q \qquad (3)$$

Here, $Pi(\varepsilon)$ is the possibility of pixels found in the *i*th box with a resolution scale of $\varepsilon$. The singularity strength $\alpha(q)$ and the multifractal function $f(\alpha)$ in relation to $\mu_i(q,\varepsilon)$ can be calculated as

$$\alpha(q) = \lim_{\varepsilon \to 0} \frac{\sum \mu_i(q,\varepsilon) ln P_i(\varepsilon)}{ln \epsilon} \qquad (4)$$

$$f(\alpha_q) = \lim_{\varepsilon \to 0} \frac{\sum \mu_i(q,\varepsilon) ln \mu_i(\varepsilon)}{ln \epsilon} \qquad (5)$$

The plot between $\alpha(q)$ and $f(q)$ is the multifractal spectrum of the image, which is computed using open-source codes ImageJ and FracLac [67].

### 2.3.3. Dynamic Mode Decomposition (DMD)

Disease progression is inherently a dynamic process. Thus, time-varying features are desirable to gauge the stage of the disease. Dynamic mode decomposition (DMD) is an image analysis algorithm that considers the system dynamic as $X'' = A*X'$, where $X'' = [A_1, A_2, A_3, ..., A_n]$ and $X' = [A_0, A_1, A_2, ..., A_{n-1}]$. Here, $A_1, ..., A_n$ are different stages of the disease (Figure 2a). For comparison purposes, proper orthogonal decomposition (POD) and principal component analysis (PCA) were also considered. The major difference between these two algorithms is that POD deals with the untreated image matrix $X = [A_0, A_1, A_2, A_3, ..., A_n]$ (Figure 2b), while PCA studies the corrected matrix $\hat{X} = (X - \overline{X})$, where each image was subtracted by the mean (Figure 2c). In comparison to POD, DMD requires an extra step, which extracts the temporal features from $X'$ to $X''$ (Figure 2c). Once the dominant features (or eigenvectors, eigenmodes) are identified, each image can be represented using a linear combination of these features. The eigenmode coefficients constitute a vector that represents the image, which can be further implemented for classification training and testing.

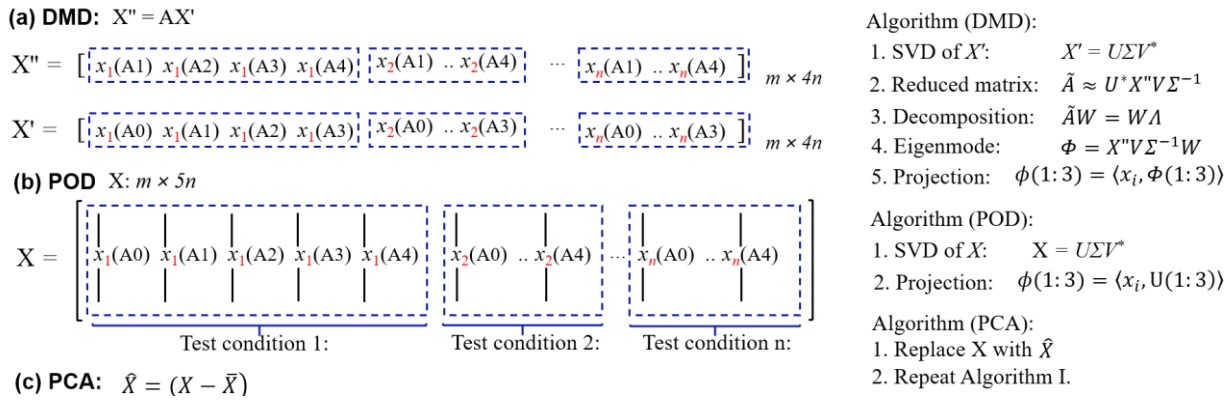

**Figure 2.** Schematic of feature extraction using different classification algorithms: (**a**) DMD, (**b**) POD, and (**c**) PCA. Each aerosol image is expressed as a single-column vector. The parameter m is the size of the image (e.g., a 600-by-600 image gives m = 360,000), and n is the number of tests with each model.

### 2.4. Image Classification of Exhaled Aerosol Images

Two classification methods, support vector machine (SVM) [68,69] and random forest (RF) [70], were used to classify healthy and abnormal lung geometries [71]. A ten-fold cross-validation method was adopted, where the data were randomly split into ten equal-sized groups. In each step, nine groups were used for training and one for testing. This step was repeated ten times. The classification accuracy is expressed below:

$$Accuracy = 1 - \frac{Total\ number\ of\ misclassified\ samples}{Total\ number\ of\ samples} \qquad (6)$$

To obtain a classification accuracy of statistical significance, the 10-fold cross-validation test was iterated one hundred times. The R package "e1071" was used to train and test the

SVM and RF models. Minitab 18 (State College, PA) was used for the calculation of the classification variability using one-way analysis of variance (ANOVA).

## 3. Results

Three test cases were presented herein. The first test case demonstrated that significant differences existed in exhaled aerosols between health and disease and among different types of diseases. The second test case demonstrated that such differences could also be measurable at different stages of one disease (e.g., a growing squamous tumor in G3). The third test case showed that even airway remodeling in deep lungs (G7-9) could be detected using the proposed fractal-machine-learning method. In Section 3.4, the third test case was further explored using DMD-extracted features, which further improved the classification accuracy.

### 3.1. Proof-of-Concept Study in Upper Airway Models

3.1.1. Image Acquisition from Physiology-Based Modeling

The AFP image is the distribution of all exhaled aerosols that are collected at the mouth opening. To develop a correlation between the AFP images and underlying lung disorders, the AFP images must be sensitive enough to the disease-induced variations in lung morphology. In Figure 3, significant differences can be observed in particle distributions among the four models, indicating a high sensitivity of the APF images to underlying airway abnormalities in the upper airways. Figure 3a,b show the expiratory airflow and exhaled aerosol distributions at the mouth opening, respectively, in the four airway geometries with one normal and three malignant lungs, i.e., with a carinal tumor, a squamous tumor at the left segmental bronchus, and constricted bronchioles [57]. The sizes of tracer nanoparticles were 0.6–1.0 μm for their low diffusivity, low inertia, and low retention rate in the airway [63,64].

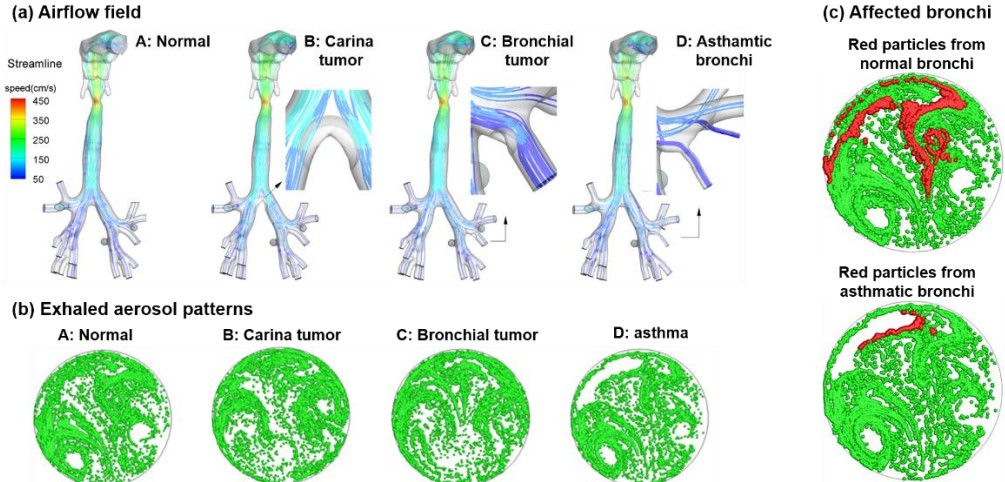

**Figure 3.** Proof-of-concept study in the upper airways: (**a**) airflow, (**b**) exhaled aerosol patterns, and (**c**) comparison of particles that were originally released from normal and constricted bronchi. One normal (Model A) and three diseased (Model B–D) were considered. The three diseases in the models B–D were carina tumor, bronchiolar tumor at G3 of the left lower lobe, and constricted bronchi in the left upper lobe (or asthma), respectively. Adapted from [57].

This hypothesis was further tested by examining the destination of aerosol particles released only from the diseased site, i.e., the two constricted bronchi in the asthma model (Figure 3c). The patterns of the red trace particles were well defined and can be used as a biomarker for asthma. Similarly, these trace particles can also be applied to localize the lung disease site and assess the severity of asthma-inflicted constrictions.

After identifying the disease site and condition, one can devise an inhalation therapy protocol to target medications at the diseased cells with a calculated dose tailored to the

disease condition. For the applications of targeted lung delivery, it is essential to know where the administered pharmaceuticals deposit within the lung. To answer this question, we first released aerosol particles from the entire mouth opening, and then selectively traced the aerosol particles that had deposited at the region of interest to their initial release locations at the mouth. Figure 4 displays the initial release locations of the aerosol particles that were deposited at the diseased sites. Conversely, if pharmaceutic agents were released only from these locations, all agents shall deposit in the diseased sites, thus maximizing the therapeutic outcomes and minimizing adverse side effects. Note that the release positions in the case of carina tumor also displayed substantial asymmetry (Figure 4b).

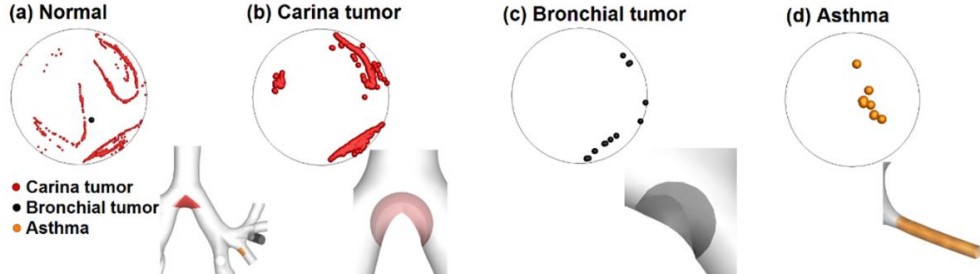

**Figure 4.** The release positions of aerosols that will deposit on the diseased regions: (**a**) normal, (**b**) carina tumor, (**c**) bronchial tumor, and (**d**) constricted bronchi (asthma). Adapted from [57].

### 3.1.2. Relative Concentration

Although the particle distributions in Figure 3b are visually distinct among models, particles that overlap each other preclude an accurate perception of particle accumulations. Figure 5 shows the particle concentration, with the blue color being zero concentration and red the maximum concentration [65].

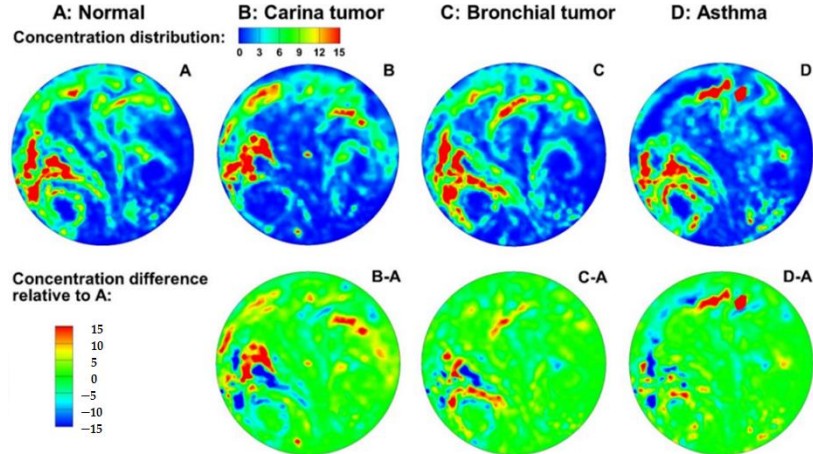

**Figure 5.** Relative concentrations of exhaled aerosols from the normal and three diseased airway models. The concentration difference in the second row was calculated by subtracting the relative concentration of each model by that of Model A. Adapted from [65].

### 3.1.3. Fractal and Multifractal Feature Extraction

Figure 6 shows the fractal dimension (FD) and the multifractal spectrum of the exhaled aerosol images shown in Figure 3. The human lung has a tree-like architecture and is a typical fractal structure [25,26] with an FD of approximately 1.57 [27,28]. During respiration, inhaled aerosols cyclically fill and empty the lung networks. It is naturally conjectured that the exhaled particles, which travel back from the deep lung to mouth by passing through the branching networks, should retain the fractal characteristics and thus are fitting for fractal analysis. Figure 6a shows the box-counting method to calculate the FD of exhaled

aerosol images from Model A (right panel), as well as the comparison of FDs among the four models for the whole image (middle panel) and a prespecified region of interest (ROI, left panel). It was observed that the whole-image FDs did not show significant differences from the normal case except Model D. Instead, significant differences were found in the regional FDs among the four models. Similar results were also observed in the multifractal spectrum (Figure 6b), where significant differences among models were absent for the whole image but prevail for the ROI. This indicated that FD in sub-regions, not the whole image, should be used for later pattern recognition and disease classification [29]. The aerosol distribution is also visualized, in Figure 6c, as a rose plot, which also exhibits large differences among the four models. It was, therefore, inferred that variations in lung structures appear to generate sufficiently large differences in the exhaled aerosols, which can be explored to correlate to the underlying lung diseases.

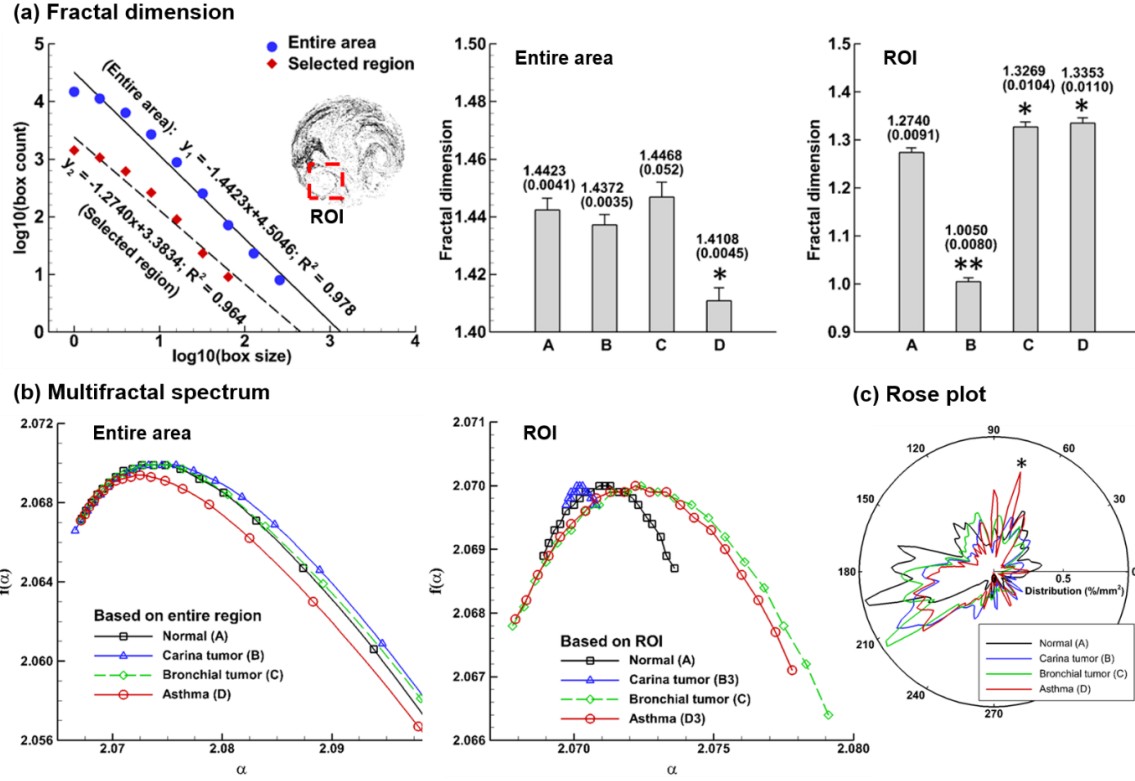

**Figure 6.** Fractal and multifractal analyses of exhaled aerosols from the normal and three diseased airway models: (**a**) box-counting method to calculate fractal dimension, FD (**left**), and FDs of the whole image (**middle**) and ROI (**right**); (**b**) multifractal spectra for the whole image (**left**) and ROI (**right**). The spatial distribution of aerosols was also visualized as a rose plot in (**c**). *: $p \leq 0.05$; **: $p \leq 0.01$.

### 3.2. Growing Bronchial Tumor

#### 3.2.1. Perturbed Airflow Field

Figure 7a shows the airflows during exhalation in each of the five stages of a squamous tumor located at a G3 branch in the left lower lobe. The airway obstruction due to the bronchial tumor noticeably modified the airflows near the tumor, as illustrated by the highly distorted stream traces. The flow perturbation was transported by the exhalation flows throughout the respiratory tract. As the tumor progressed from case A to E, more airway obstruction and higher flow resistance were expected, which caused a reduced volumetric flow rate under similar respiration efforts. Due to airway obstructions, the trajectories of aerosol particles were also disturbed, which could perceivably change the exhaled aerosol distributions. Figure 7b compares the velocity contours at three cross-sections (a–a', b–b', c–c') among the five stages of the growing tumor. As expected, the

flow speed progressively dropped with increasing tumor size. As the expiratory airflow travelled to the mouth, the difference in airflows gradually decreased due to the mingling of flows joining from neighboring branches. It is reminded that the aerosol profile is determined by both local airflows and the flow histories. Although the flow fields in the trachea looked similar among the five models, the aerosol patterns could be highly distinct because of their time-integrative properties.

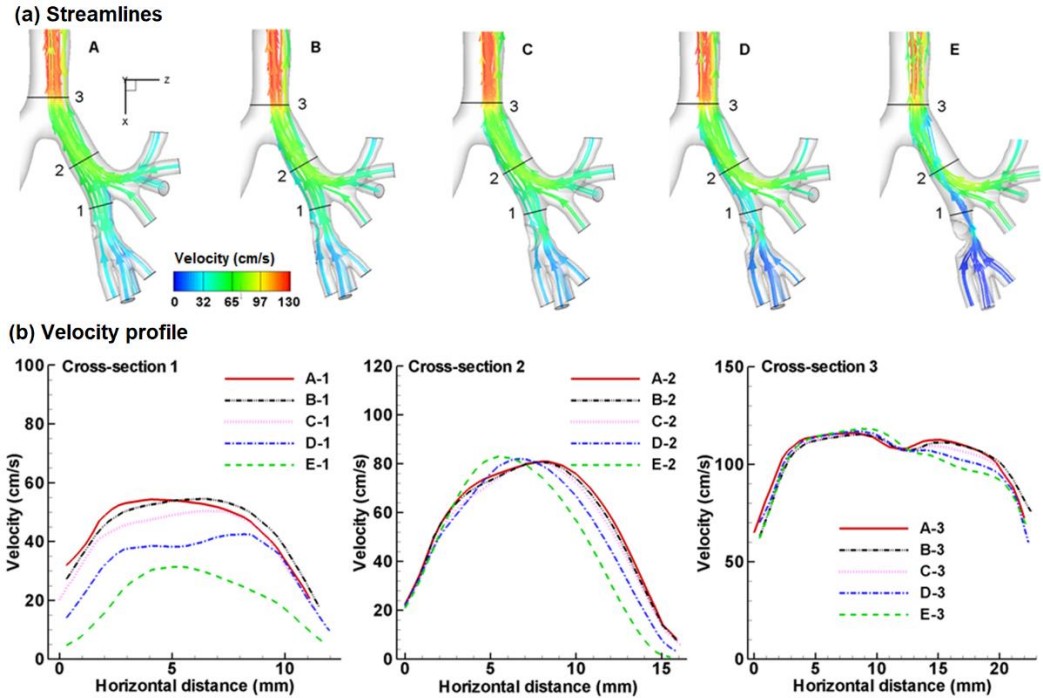

**Figure 7.** Test case 2: airflow fields at the five stages (A–E) of a developing tumor (**a**) stream traces and (**b**) cross-sectional velocity profiles.

### 3.2.2. Patterns of Exhaled Aerosol Fingerprints

Distributions of the tracer particles collected at the mouth opening are visualized in Figure 8a. Both differences and similarities were observed in particle distributions among the five stages of the bronchial tumor. Among many differences in Figure 8a, two are noteworthy. First, a recirculation zone was observed in the left lower corner that occurred at all stages of the tumor, while a smaller recirculation zone in the right corner shrank in size gradually from case A to E (Figure 8a). These two recirculation zones were not symmetric relative to the centerline of the circular filter, which was presumably due to the left-right asymmetry of the lungs. Second, large differences existed in the particle distributions between the two regions outlined by a dashed red circle and a box (Figure 8a). In both regions, the particle concentration decreased with increasing tumor size. Moreover, increasingly scattered particle distributions were observed in the dashed red box with increasing tumor size from A to E.

Figure 8b shows the relative concentrations (or local density) of exhaled particles on the mouth filter, with zero concentration in blue and maximum concentration in red. Compared to particle profiles (Figure 8a), the concentration image (Figure 8b) identified the peak particle accumulations (red color), which could not be identified in the exhaled aerosol images per se due to particle overlapping. In this test case, both recirculation zones (i.e., at the left and lower-right corners) exhibited elevated levels of aerosol accumulations. The accumulation levels in left recirculation remained relatively unchanged for all stages herein, while that in the lower-right corner declined continuously with the tumor growth. A similar decline can be observed in Figure 8a in the right-middle zone (red rectangle).

These time-varying correlations can be utilized as biomarkers to probe the progression of the tumor, which is further discussed in later sections.

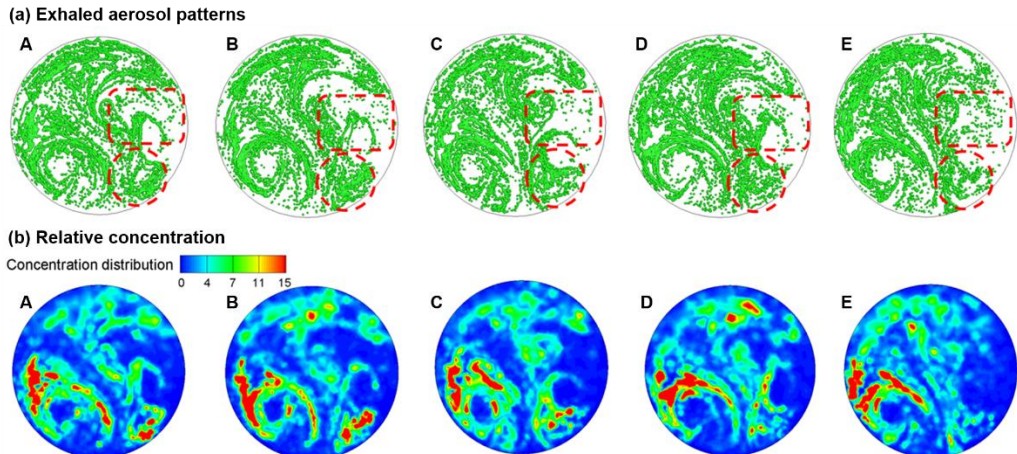

**Figure 8.** Test case 2: comparison of (**a**) exhaled aerosol patterns and (**b**) relative concentration among the five stages (A–E) of a growing squamous tumor.

### 3.2.3. Fractal Dimension

Fractal dimensions (FD) were computed for exhaled aerosols images both in the whole image and specified ROIs (Figure 9). In each case, the standard deviations in FD were computed from aerosol images that were predicted based on stochastically generated inlet particle profiles ($n = 5$). A consistent FD variation with growing tumor sizes was not observed for the whole images, as indicated by the erratic change of the FD values in Figure 9a. Rather, a constant decrease in FD was noted for the two ROIs with tumor growth. This finding confirmed the previous observation that malignant lungs gave lower FDs than those in normal lungs [72]. Different from the small FD variations for the whole images, large variations in the FD were predicted in both ROIs across the five disease stages. Moreover, the derivation of FD from the normal (case A) increased steadily in diseased models from B to E. Accordingly, the ROI-based FD seem to be a feasible biomarker to monitor tumor growth and gauge disease severity. Considering that FD in sub-regions may carry more concentrated information about the disease, each aerosol image was further split into a $6 \times 6$ grid. Figure 9b shows the FD values in each grid. As expected, large differences in color patterns were observed across the five tumor stages, with a unique pattern in each stage.

### 3.2.4. Multifractal Spectrum Analysis

The multifractal spectra of the exhaled AFP images are shown in Figure 10 for the five stages of the growing tumor. Interesting transition patterns can be observed across stages (Figure 10a), as illustrated in the 3D surface plots of particle localization at the mouth. First, the height of the peaks in the left corner (solid black arrow) decreased continuously from Model A to E. Second, the aerosol concentration in the left-middle zone (red rectangle) decreased as the tumor grew from Model A to E. These two outlined zones were used as the ROIs for multifractal spectrum analyses, as shown in Figure 10b,c, respectively.

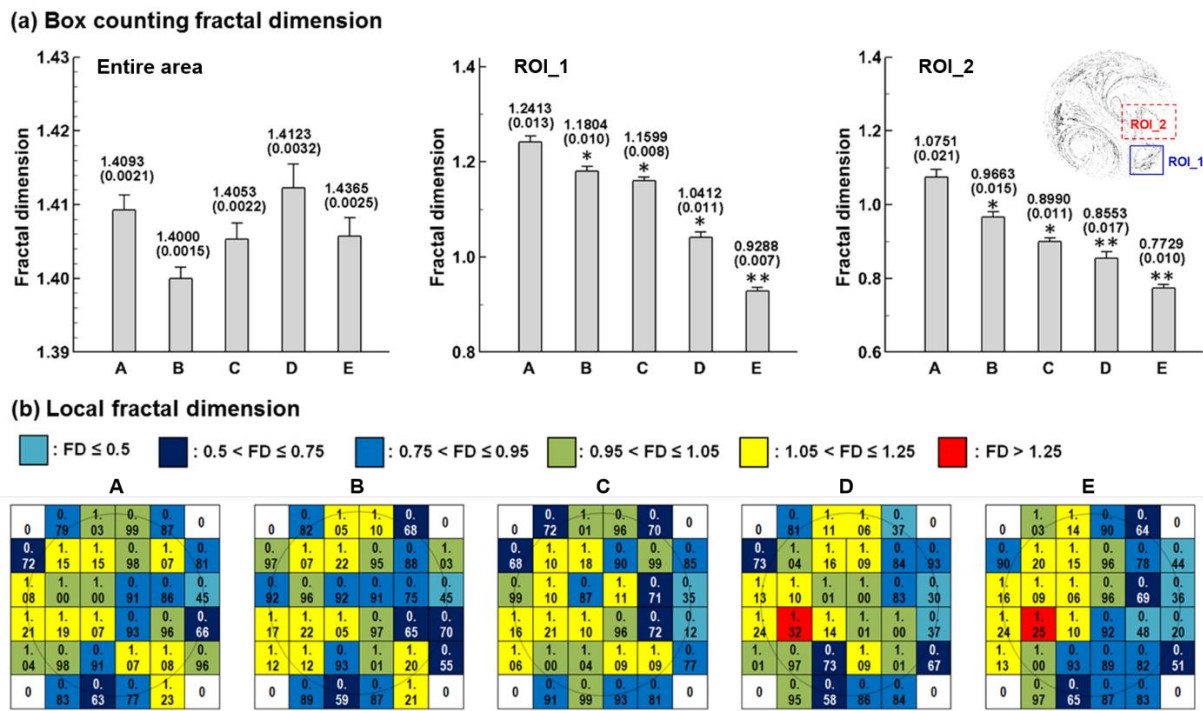

**Figure 9.** Fractal analysis in the test case 2: (**a**) comparison of fractal dimensions among the five tumor stages for the entire image (**left**), ROI_1 (**middle**), and ROI_2 (**right**); (**b**) FD distribution by dividing the image into a 6 × 6 lattice for the five tumor stages. *: $p \leq 0.05$; **: $p \leq 0.01$.

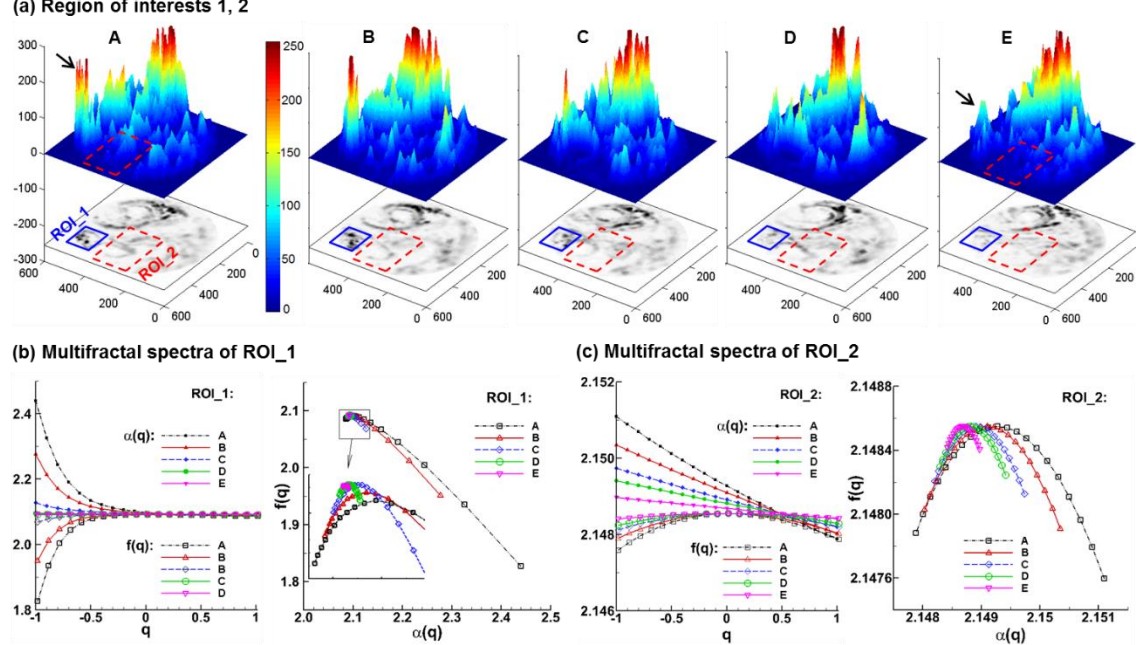

**Figure 10.** Multifractal spectrum analysis in the test case 2: (**a**) 3D plot of aerosol concentrations; (**b**) comparison of the multifractal spectra in ROI_1 across the five tumor stages (A–E); (**c**) comparison of the multifractal spectra in ROI_2 across the five tumor stages (A–E).

Again, different profiles among the five disease stages were observed in the multifractal spectra in both ROIs (Figure 10b,c), lending support to the idea that the differences in exhaled aerosols can be significant and are quantitatively comparable between health and disease. Differences between the two ROIs were also predicted in their multifractal

spectra. As illustrated in Figure 10b, the f($\alpha$)-spectra in the first ROI converged to a single point (q = 0), which was not observed in the second ROI (Figure 10c). As a result, the first ROI is a mono-fractal, while the second ROI_2 is considered to be multifractal. Comparing the right panels in Figure 10b,c, the discrepancy ($\alpha$max − $\alpha$min) was associated with the heterogeneity of the aerosol images. This discrepancy decreased persistently for both ROIs considered as the tumor grew in size from Model A to E, indicating a decrease of pattern heterogeneity with tumor growth. This result corroborates the reports in the literature [73,74] that a narrower range of $\alpha$ often gives a lower value of lacunarity, which is an alternative indicator of image heterogeneity. Similar examples include vascular networks with high density [74] and soils with low porosity [73].

### 3.3. Asthmatic Bronchioles in Small Airways

### 3.3.1. Airflow Field and Exhaled Aerosol Images

Figure 11 compares the exhalation flows and aerosol distributions among the five stages of an asthmatic bronchiole at G7. The bronchiolar constriction significantly altered the velocity field in the bronchiole (Figure 11a). These flow perturbations were transported by exhalation flows downstream to approximately four bifurcations beyond the constricted bronchioles. Disturbances to solid particles persisted even a longer distance because of their inertia. The constricted bronchioles also increased the flow resistance and decreased the volumetric flow rate through it, which further modified the exhaled aerosol behaviors. It is emphasized that although downstream airflows appeared similar, the aerosol distributions could differ significantly due to upstream particle perturbations. This was evident in the strikingly distinct patterns of exhaled trace particles that were originated from branches of five constriction levels (Figure 11b).

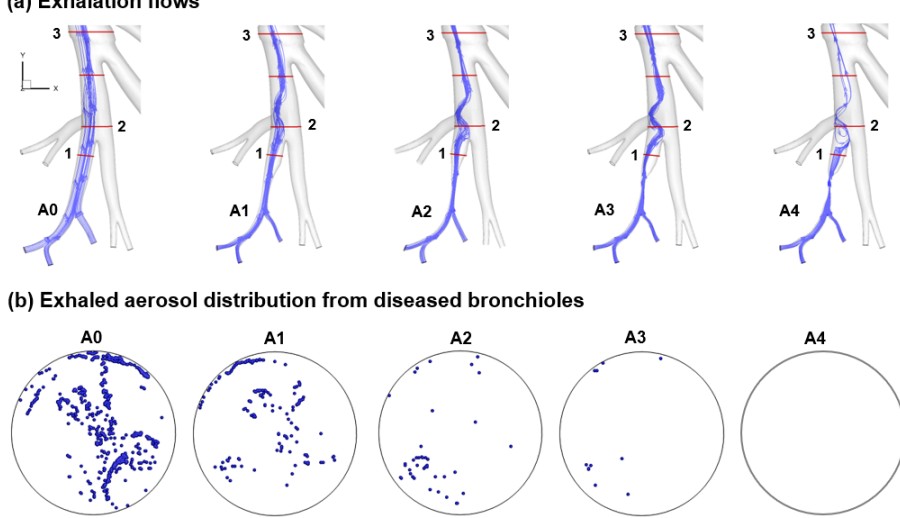

**Figure 11.** Airflow and aerosol dynamics in the test case 3: (**a**) comparison of streamlines near the constricted bronchioles at G7 among five stages and (**b**) exhaled particle distribution for particles originated only from the disease-affected bronchioles at varying stages.

### 3.3.2. Fractal-Feature Extraction

To vectorize the exhaled AFPs, the aerosol image shown in Figure 12a was split into an n × n matrix, and the FD in each grid was computed. Figure 12b shows the FDs at a 3 × 3 matrix while Figure 12c shows the FDs at 6 × 6. The color code was calculated as the ratio $\alpha$(i) = FD(i)/FD(A0), i = A0, A1, A2, A3, A4, with the red being the highest ratio while the dark blue the lowest. The pattern of color arrays was observed to be distinct for each model and at each matrix resolution (n × n). Each image was expressed as a vector consisting of n features by stacking the n × n FD values in one row. The distinct

color patterns between different matrix resolutions for the same image can impart more information in revealing the subtle variations that are hidden in the images.

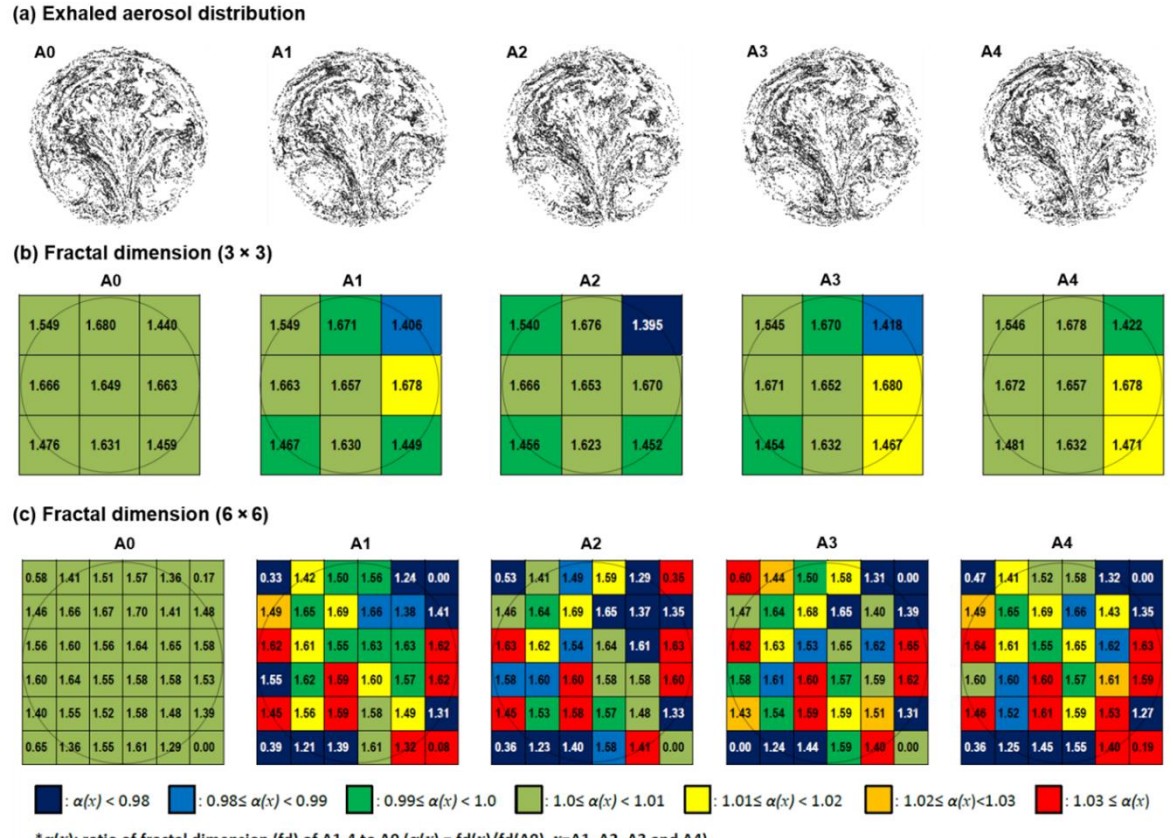

**Figure 12.** Exhaled aerosol patterns and FD-based feature extraction in the test case 3: (**a**) exhaled AFPs for the five stages of the bronchiolar tumor, (**b**) FD distribution at 3 × 3 resolution, and (**c**) FD distribution at 6 × 6 resolution.

### 3.3.3. Database Quality Check

Principle component analysis (PCA) was carried out to check the dataset quality, as shown in Figure 13. It was observed that the first principal component (PC1) had an eigenvalue of 15.5 and captures 20.6% of the total data variance (Figure 13a). The PC2 has an eigenvalue of 10.5 and captured 14.0% of the data variance. In combination, these two leasing PCs captured 34.6% of the data variance. The first five PCs captured 56.8%, whereas the other 85 PCs captured the remaining variance of 43.2% (100 PCs retained in this case). As a result, unimportant components could be safely neglected without noticeably compromising classification accuracies. To identify outliers in the image database, the Mahalanobis distances were plotted (Figure 13b) for all images in the database [75]. No obvious outlier was found in this database because all data points fell below 10.65, the reference line for outlier existence [75].

### 3.3.4. Classification Using SVM and Random Forest (RF) Algorithms

Two classification methods, SVM and random forest (RF), were applied to train and test the image data at varying matrix resolutions. Figure 14a,b show the statistical results of the five-class classification (A0–4) using the random forest (RF) and SVM, respectively. As expected, the lowest classification accuracy was predicted at the 3 × 3 resolution because of its low sampling resolution. The classification accuracy improved with increasing sampling resolutions until reaching the optimal one at 12 × 12 and dropping thereafter as the sampling resolution became even finer (Figure 14a,b). This accuracy deterioration may be due to the probability that a too-small sampling grid may miss the disturbance

signals from the small airways. The highest accuracy with SVM was also found at 12 × 12. Overall, RF persistently outperformed SVM in this test case (Figure 14a vs. Figure 14b).

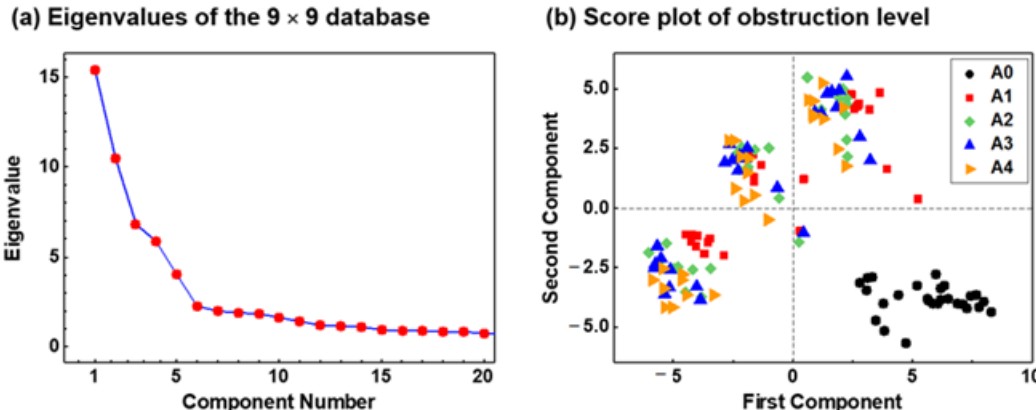

**Figure 13.** PCA-based quality check of images in the test case 3 that was mapped in a 9 × 9 matrix: (**a**) eigenvalues of the first 20 features, (**b**) score plot of the airway obstruction level vs. the first two features (principal components).

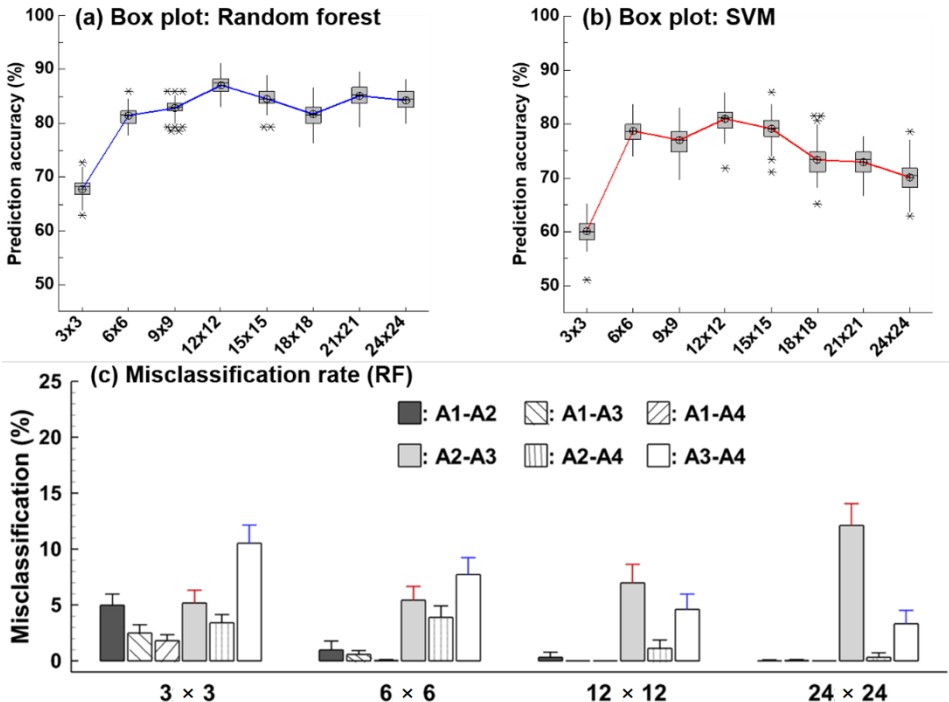

**Figure 14.** Image classification in the test case 3: (**a**) the predicted five-class classification accuracies based on RF at different sampling resolutions, (**b**) those based on SVM, and (**c**) the misclassification rate based on RF. The asterisks in (**a**,**b**) are outliers.

### 3.3.5. Misclassification Analysis

Figure 14c shows the misclassification rates using RF in this test case. The highest rates of misclassification were observed for the pairs of A2–A3 and A3–A4, indicating its strong association with structural similarities. Conversely, misclassifications were much reduced for A2–A4, whose geometrical difference is more pronounced than that of A2–A3 or A3–A4. Regarding different resolutions, extensive misclassifications were found at either highly coarse (3 × 3) or highly fine (24 × 24) resolutions. Misclassification spikes were absent at 6 × 6 and 12 × 12, although there were still misclassifications for A2–A3 and A3–A4 at these two resolutions (Figure 14c). It was conjectured that the best classification accuracy at 12 × 12 can result from the disease site at G7 that has 128 (27) sister bronchioles and,

if being evenly mapped, needs an 11.3 × 11.3 matrix. However, the verification of this hypothesis needs future investigations.

### 3.4. Dynamic Mode Decomposition (DMD) to Catch Disease Growth

#### 3.4.1. DMD vs. Conventional Algorithms

In this section, we explain the usage of dynamic mode decomposition (DMD) to extract dominant features of the images and demonstrate its superiority over conventional methods, such as fractal dimension, POD, and PCA. As a quick explanation, both POD and PCA are rooted in the singular value decomposition (SVD) that can handle non-square matrices. POD decomposes the system into mutually orthogonal eigenmodes that are spatially independent of each other. PCA is similar to POD but removes the mean to emphasize the image contrast. PCA has been widely applied in machine learning for dimensionality reduction and feature extraction. DMD is also an SVD-based decomposition method, such as POD and PCA. However, the DMD-modes directly capture the temporal evolution of the system and are inherently suitable to study time-varying systems, which develop around an attractor (e.g., one type of lung disease) with transient oscillators (e.g., diseases at different stages). Details of the mathematical algorithms for DMD, POD, and PCA can be found in Xi and Zhao [76].

#### 3.4.2. DMD Feature Extraction of Exhaled AFP Images

Figure 15a displays DMD-predicted dynamic eigenvalues in the complex plane, with each dot representing one eigenmode. Most modes were observed to be within the unit circle, while some eigenmodes were laying on or close to the right side of the circle. The radius of each mode (i.e., distance from the center) denotes its dynamic characteristics, growing if the radius is larger than one, while decaying if the radius is less than one. In Figure 15a, nearly all eigenvalues fall within the circle, which signifies a strong damping and a stable system in this test case.

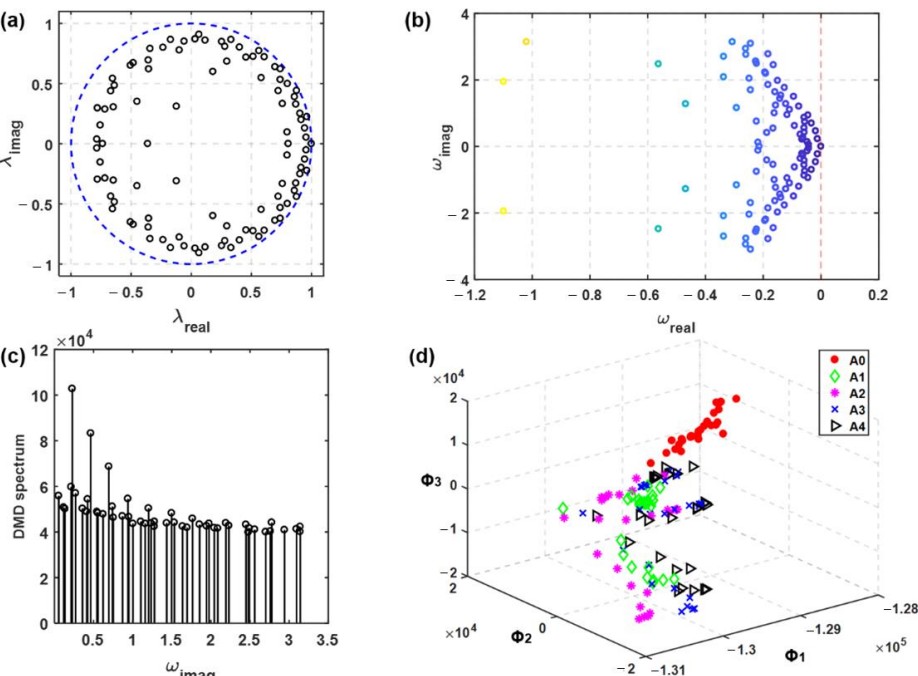

**Figure 15.** DMD analysis of exhaled aerosol images in the test case 3: (**a**) DMD eigenvalues $\lambda$, (**b**) transformed eigenvalues $\omega = \log(\lambda)/(2\pi)$, (**c**) DMD energy spectrum, and (**d**) eigenspace spanned by the first three DMD eigenvectors.

The transformed eigenvalues are shown in Figure 15b, expressed as $\omega = \log(\lambda)/(2\pi)$. The real part of $\omega i$ in each point represents the response of the eigenmode to inputs

(e.g., disease types and stages). Similarly as in Figure 15a, while decay occurred for most modes, several modes laid on or in the proximity of the red line, suggesting neutral stability and slow decay for these modes. Figure 15c shows the DMD spectrum. Three spikes were noted at frequencies f0, f1, and f2. It was no surprise that f0 was related to the respiration flow rate (repeated every three cases), f1 to the disease stage (repeated every five cases), and f2 to the diameter of tracer particles (repeated every nine cases). It was noted that that three inhalation rates and nine particle sizes were considered in this test case.

The test images were subsequently projected into the eigenspace spanned by the leading eigenmodes. Figure 15d shows the images in the eigenspace spanned by the first three modes. Reasonable clustering and separation of data points were observed according to the disease stage. It is emphasized that eigenvalues from DMD are complex numbers, with the real part signifying the growth/decay and the imaginary part signifying the oscillation. This capacity to characterize both the growth/decay and oscillation behaviors is highly advantageous when studying temporal signals and progressive systems.

### 3.4.3. SVM and RF Classification Based on DMD Features

The classification accuracies for five stages of an asthmatic bronchiole (A0-4) at G7 are shown in Figure 16. This figure was plotted from the 10-fold cross-validation tests that were iterated 100 times. To evaluate the effects of extracted features, one retained 25 eigenmodes and the other retained 100 (Figure 16a vs. Figure 16b). To study the effect of classification algorithms, SVM and RF were considered. Overall, RF outperformed SVM for all tests. When using the RF classification algorithm, the classification accuracy using the DMD-extracted feature was significantly higher than that obtained using POD- or PCA-extracted features (Figure 16a,b, upper panel). However, when using the SVM, an insignificant discrepancy was observed in the classification accuracy among the three features (Figure 16a,b, lower panel). It was also observed that retaining more features could increase classification uncertainty, as indicated by the outliers presented in the 100-mode case (Figure 16b) versus no outlier with 25 modes (Figure 16a). This uncertainty might be caused by noisy data and associated features that contaminated the disease stage-related features.

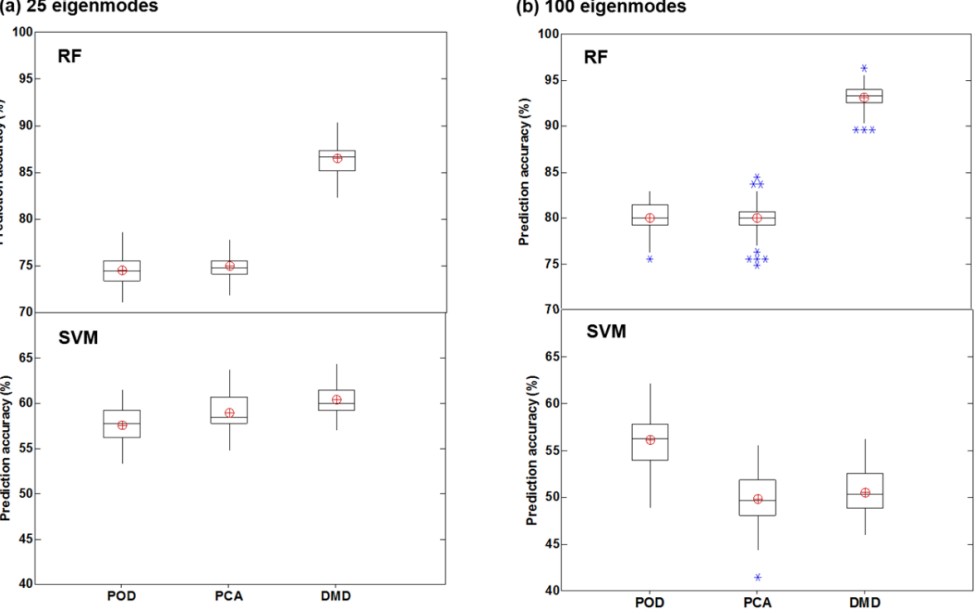

**Figure 16.** Classification accuracy of the five stages of an asthmatic bronchiole (A0-4) by retaining a varying amount of eigenmodes: (**a**) 25 and (**b**) 100. Various combinations between feature extraction (POD, PCA, DMD) and classification (SVM and RF) algorithms were compared. The asterisks in (**b**) are outliers.

## 4. Discussion

In many circumstances, it is nontrivial to find a subset of features that can effectively characterize the system of interest. In this review, we introduced a non-invasive method for lung disease diagnosis based on exhaled aerosols and presented the latest developments in extracting features from the exhaled aerosols. The performance of extracted features in correlating to underlying respiratory obstructive diseases was compared in terms of classification accuracy in both the tracheobronchial region (G3) and small airways (G7). It was found that the DMD-extracted features, in combination with the RF classification algorithm, gave the highest prediction accuracy.

### 4.1. DMD vs. Other Feature Extraction Algorithms

Why DMD can extract more informative features and give improved classification accuracy compared with POD or PCA is discussed below. DMD considers the time-varying features related to different stages of a squamous tumor and an asthmatic bronchiole and, therefore, captures the phase transition from one stage to another. Instead, POD and PCA extract mutually orthogonal eigenmodes from the image matrix; loss of information associated with the disease growth often occurs during the averaging process to construct the correlation matrix. Furthermore, the POD- and PCA-extracted eigenmodes are not temporally independent of each other [77], which would also adversely affect their classification accuracies.

The DMD-RF method outperformed the fractal-RF method by a large margin (7.8%). Fractal dimensions have been explored in previous studies [56,78] to vectorize exhaled aerosol images at different sampling resolutions. The best fractal-RF classification accuracy for the five-class classification of an asthmatic bronchiole was 87.0% [56]. Despite the ability of fractals in quantifying complex patterns, information losses are inevitable due to the box-counting principle underlying the fractal dimensions. It is not clear yet how the information loss impacts the disease classification. In contrast, DMD-based features (eigenmodes) are directly extracted from raw images using SVD to find the leading coherent structures, thus minimizing information loss. The second DMD-based enhancement comes from the temporal dynamics that are deeply rooted in disease growth. Time-related features can be disclosed that can be otherwise overlooked using static decomposition approaches such as POD and PCA.

Detecting diseases in small airways is more challenging than in the central and large airways owing to their much weaker perturbation signals. Two factors can contribute to these weak signals: small perturbations due to small airway sizes in deep lungs and a longer period of signal decay during expiration. It is critical that these signals can be captured as exhaled aerosol fingerprints at the mouth opening and can be reversely traced back to the origin of these signals. In this review, we demonstrated that the DMD-RF method was sensitive enough to detect structural variations in small airways of 2 mm diameter or less in AFP images. In the test case of the asthmatic bronchiole with a diameter less than 1.87 mm, the classification accuracy was 94.8% among five disease stages when using 100 features. In addition, this asthmatic bronchiole model has only one bronchiole deformed and is expected to generate much weaker disturbance signals relative to real-life asthma, where often a large area is affected and thus is more suitable for diagnosis. Note that the high accuracy hereof may result from many idealized assumptions, as will be detailed in 4.3; the classification accuracy is expected to be lower in clinical settings due to numerous compounding factors.

### 4.2. Future Directions

Convolutional neural network (CNN) has gained ever-increasing popularity in recent years with mounting evidence of improved performance over conventional machine learning methods [79,80]. One major advantage of CNN is that it can automatically learn the features from the training images and thus can directly use the input images. In addition, the rich features that CNN can learn at multiple levels have given rise to many successful

applications in medical image analysis [81,82]. However, there are specific challenges in using CNN models. Due to its self-learning nature, a CNN model needs a large dataset to capture the image features effectively; however, quality medical images are usually not readily available. In this test case, a dataset of 405 exhaled aerosol images was used that might be enough for classifications with SVM and RF but was still far from adequate for meaningful CNN training and testing. Future assessment of CNN performance in automatic feature extraction and classification of exhaled aerosol images is warranted as more AFP images become available.

In this review, local bronchiolar constrictions in G3 (a growing tumor) and G7 (asthmatic bronchioles) in the left lung were selected to demonstrate the applicability of exhaled AFP-based breath tests in lung diagnosis. It was noted that disease-induced airway remodeling can happen in the respiratory tract anywhere and at multiple places [83–86]. Will the overlaying of particle perturbations from different disease sites make it too complex to differentiate by the proposed fractal-RF or DMD-RF methods? Our best guess is that it will not. All AFP images, regardless of how complicated they appear, can simply be characterized as feature vectors for classification purposes. As different types of diseases will give rise to their unique patterns of exhaled aerosols, a database of common respiratory obstructive diseases can be developed to train and validate a generic diagnostic system. The database can readily be augmented by including new AFP images. As more AFP images are becoming available, the existing diagnostic system can be constantly refined for improved accuracies.

*4.3. Assumptions and Limitations*

The high classification accuracy in this study may result from idealized assumptions and controlled airway structural remodeling. Results in this study do not apply to other respiratory diseases, nor can they be directly compared to the diagnostic rate of current diagnostic techniques. Many uncertainties exist that can complicate the training process and reduce classification accuracies. Patient-related uncertainties include the body position, inhalation rate, mouth shape, and tongue position. The lung geometry itself can also differ among patients. The compounding influences from certain factors can be alleviated by careful planning; other factors demand additional investigations. For example, using a mouthpiece is anticipated to minimize the impacts of the oral cavity shapes and tongue positions [87,88]. Likewise, standardization of breathing and sitting posture should mitigate compounding influences from the variability in respiration rates and body positions, respectively.

Several assumptions may restrict the physical realism of the results presented in this review, such as steady inhalation and exhalation, noncompliant airway walls, uniform outlet pressure, and a small cohort of respiratory disease models. Tidal breathing [38] and compliant walls [89] can exert noticeable impacts on both airflow and aerosol dynamics. Moreover, the ventilation distribution can differ between the case with a uniform outlet pressure and the case with specific resistance and compliance of downstream airways [90,91]. It is acknowledged that uniform pressures were adopted at the outlets in all test scenarios in this study due to the lack of resistance/compliance data. The impact of this assumption on diagnostic accuracy, however, should be minor, considering the consistency of this assumption for all images herein, as well as the fact that classification was based not on individual patterns, but on similarities/differences among patterns. Considering the high variability in site, size, and type of respiratory diseases, detecting a disease that has not been included in the database is impractical. One advantage of the simulation-based database is that it can be readily extended by simulating more scenarios of obstructive diseases. The development of the training database could be an ongoing effort, with new sample images helping improve both the prediction accuracy and statistical power. Considering the respiratory airway models, only two lung geometries were considered, while intersubject variability was neglected [92]. Moreover, only numerical simulations

were undertaken. Complementary in vivo and in vitro studies are required to verify the physiology-based modeling and predictions herein.

*4.4. Potential Roadblocks and Solutions*

There are some frequently asked questions regarding the clinical applications of this proposed aerosol breath testing, with the best-practice approaches to address these roadblocks from our perspective.

4.4.1. How the Proposed Breath Test Can Be Implemented in Clinical Settings?

The patient will inhale a bolus of particles at a slow and steady inhalation speed, then exhale. A mouth filter will be used to collect the exhaled aerosols, which deposit in the filter surface and form a unique pattern. This filtered image will then be analyzed to determine the chance of a specific respiratory disease.

4.4.2. How Can a Classifier Be Developed when There Is No Record of Aerosol Images at the Patient's First Visit?

The proposed breath test is envisioned to have two steps (i.e., screening and validation). In the first step for screening purposes, the exhaled aerosol image collected from the patient will be tested against a population-based classifier that had been trained on the database of a specific disease (such as COPD), to determine the probability of the disease in this patient. If the probability is high, follow-up breath tests are needed for validation purposes. In the second step, the aerosol images collected thenceforth will be grouped into a new database and used to train a personalized classifier to verify the initial screening result. Because of the persistence (or progression) of the disease, common features (or feature evolution) will show up in the time sequence of aerosol images. As such, this personalized database can also be used to measure disease progression or treatment efficacy of the patient. Moreover, this database can be incorporated into the population database to improve the applicability of the population-based classifier.

4.4.3. There Is Significant Intersubject Variability in Upper Airway Morphology and Breathing Habit. How Can These Compounding Effects Be Minimized?

Effects from these factors can be minimized through standardization. For instance, adopting a mouthpiece during the test will minimize the impact of the shape of the mouth and tongue position. Likewise, standardizing the patient's breathing pattern (slow and steady) and sitting position (upright) can reduce the complication from breathing and body position.

4.4.4. What Effects on the Breath Test Results Are Expected from Turbulent Flows?

The turbulent flow will reduce the differentiability of the exhaled aerosol images, and thus may reduce the prediction accuracy of the classifier. However, the method proposed in this study doesn't depend on the behaviors of individual particles, but rather on the patterns of particle distributions, which will be different between two different lung structures. Moreover, slow and steady inhalation will be used to minimize turbulence effects.

4.4.5. Diseases Can Occur Anywhere in the Lung. How to Tell the Location of the Disease from an Aerosol Image?

That is why we need a large database and use machine learning to analyze the images. Healthy lungs share a tree-like architecture while a given phenotype of diseases (like COPD) share similarities in airway remodeling. At the same time, exhaled aerosols from different sites of the lungs with different severities will exhibit distinct patterns. In other words, these aerosol images contain information of both similarities and differences; machine learning can be used to extract those features of interest and correlate them to the disease site and severity. Given a large enough database, there will be enough features to differentiate a wide spectrum of lung diseases.

## 5. Conclusions

In conclusion, nanoparticle-based exhaled aerosols from human lungs exhibit unique features, and a deviation may be associated with underlying structural variations in the lungs. Different algorithms of feature extraction were discussed, and their capacity in characterizing airway variations was evaluated in both larger and small airways of anatomically accurate lung models. The fractal dimensions of the exhaled aerosol images at multiple resolutions appeared to effectively capture the geometrical variation that accompanies the progression of respiratory diseases. Significantly improved performances were achieved using dynamic algorithms for feature extraction (DMD) than with static algorithms (fractal, POD, and PCA). Considering machine learning models, RF consistently outperformed SVM for both static and dynamic features considered in this study.

**Funding:** This research received no external funding.

**Acknowledgments:** William Zouzas is gratefully acknowledged for reviewing this manuscript.

**Conflicts of Interest:** The authors declare no conflict of interest.

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
