# Peer review of "Deciphering Exhaled Aerosol Fingerprints for Early Diagnosis and Personalized Therapeutics of Obstructive Respiratory Diseases in Small Airways"

_jnt, doi:10.3390/jnt2030007_

Round 1

Reviewer 1 Report

Congratulations to the authors for this excellent article.

I find it basically flawless, perfectly written and full of details, including some FAQs.

I have only two comments:

-Please add more details in methods section about the patient whose MRI gave birth to the lung models. Presence of any pathology, smoking or not, race and bmi. 

-I found some small english grammar errors thoroughout the manuscript. Please have a re-check.

Author Response

Please add more details in methods section about the patient whose MRI gave birth to the lung models. Presence of any pathology, smoking or not, race and bmi. 

Response:

  1. More details were given in the methods of the reconstruction of lung models (lines 148-155): “Mimics (Materialise, Ann Arbor, MI) were used to segment the airspace from other organs. Due to the presence of artifacts in the segmented geometry, the polylines that enclosed it were extracted, which were further used as the scaffold to reconstruct the airway surface geometry by patching it with 3~5-edge faces. This method allowed the cleansing of apparent artifacts, an improved computational quality, and a controlled modification of regional airway structures. The downside of this mothed was time-consuming and labor extensive, as most of the procedures, such as artifact-cleansing and surface-patching, were manual.”
  2. The subject did not smoke and didn’t have known respiratory disease at the time of scanning. His race and BMI were unknown.  The sentence on lines 148-149 was revised as "The first model was developed based on MRI scans of a 53-year-old non-smoking male with no respiratory disease.”

I found some small English grammar errors throughout the manuscript. Please have a re-check

Response: We have carefully checked the grammar throughout the manuscript and corrected several places (line 60: add ‘and’; line 261: change ‘f’ to ‘of’; line 598: add ‘the’; line 708: change ‘impact from’ to ‘impact of’; line 713: add ‘The’).

Reviewer 2 Report

The article establishes respiratory tract models of different states and simulates the state of aerosol exhalation under different models to obtain the aerosol profile (AFP). Then, through the analysis of AFP, it explores the influence of the disease location and the disease development state on AFP. This article explores the changes in AFP and feature values ​​under two conditions, including the occurrence of diseases at different locations, and the same disease location at different disease periods, by using the feature extraction methods of relative concentration, fractal degree, and multifractal spectrum. By combining the feature extraction algorithms of DMD, POD, and PCA with the classification algorithms of RF and SVM, the article analyzes AFP and explores the possibility of using AFP to detect the course of asthma at G7. It can be seen from the article that the research team has rich research experience in the corresponding field. Of course, during the review process, I have the following suggestions and questions, and I hope to discuss further:

  1. First, there are several textual errors in the article. The first is the second page, line 60. There should be a comma missing between tuberculosis and rheumatoid arthritis. The second is the sixth page, Formula 6 explains the accuracy. The total number of numerator part is labeled as total number f.
  2. The image richness of the article are impressive, but the resolution of the image is poor. When reading, the text and graphic information are not displayed clearly, which brings great obstacles to reading.
  3. The results in section 3.1 show that different diseased parts can change the APF image, which can also be reflected in the extracted features. My question is, if only AFP is known, can the diseased part be deduced from the APF image, what method is used, and what is the criterion?
  4. Page 20, lines 691-693, the text says, "In the second step, the aerosol images collected thenceforth will be grouped into a new database and used to train a personalized classifier to verify the initial screening result." I am more confused about how to divide an unrecorded patient into a new database to build a personalized classifier? This classification model is a typical supervised learning algorithm. The prerequisite for building a personalized classifier is that a clear disease label can be learned by the algorithm. But since the disease label of the patient is known, why do we need to build a personalized model?
  5. The article obtained the corresponding data through simulation. We know that there may be a certain deviation between the simulated data and the actual data. For the performance of the method proposed in the article under actual conditions, further research may be needed. At the same time, the feature extraction algorithm will definitely lose features. Therefore, the method of directly using the original image for classification and recognition needs to be further explored and compared with the existing methods. Of course, because the CNN algorithm has some requirements for the amount of data, as your article said, these still need to enrich the database and further research.

Author Response

  1. First, there are several textual errors in the article. The first is the second page, line 60. There should be a comma missing between tuberculosis and rheumatoid arthritis. The second is the sixth page, Formula 6 explains the accuracy. The total number of numerator part is labeled as total number f.

Response: We thank the Reviewer for the careful review.  These two errors were corrected: (line 60: “tuberculosis and rheumatoid arthritis”, Formula 6 (between lines 260 and 261): changed “f” to “of”.

  1. The image richness of the article are impressive, but the resolution of the image is poor. When reading, the text and graphic information are not displayed clearly, which brings great obstacles to reading.

Response: The original images were tiff files with 300dpi resolution.

  1. The results in section 3.1 show that different diseased parts can change the APF image, which can also be reflected in the extracted features. My question is, if only AFP is known, can the diseased part be deduced from the APF image, what method is used, and what is the criterion?

Response: That's a great question related to how this method works. Briefly, if an AFP is known, this AFP will be tested in a machine-learning-based model that has been developed on an existing database. This will provide the preliminary diagnosis and further tests will be conducted for validation and progression monitoring.  Once a database of images was obtained, a personalized machine-learning-based model can be developed.  As the temporal variation of the image is more relevant to the disease progression, the new model is expected to provide a more accurate diagnosis.  This model can be further improved with more images. A similar description can be found in Section 4.5.2. 

  1. Page 20, lines 691-693, the text says, "In the second step, the aerosol images collected thenceforth will be grouped into a new database and used to train a personalized classifier to verify the initial screening result." I am more confused about how to divide an unrecorded patient into a new database to build a personalized classifier? This classification model is a typical supervised learning algorithm. The prerequisite for building a personalized classifier is that a clear disease label can be learned by the algorithm. But since the disease label of the patient is known, why do we need to build a personalized model?

Response: The second, personalized classifier will be built on the variations of the AFP images, which are expected to be reflective of the disease progression. The disease labels (severity level hereof) can be acquired from the preliminary diagnosis (the first clarifier) and/or other diagnostic modalities such as CT, which can be used to train the personalized classifier.  Future images can then be tested in the personalized classifier for prognosis.     

  1. The article obtained the corresponding data through simulation. We know that there may be a certain deviation between the simulated data and the actual data. For the performance of the method proposed in the article under actual conditions, further research may be needed. At the same time, the feature extraction algorithm will definitely lose features. Therefore, the method of directly using the original image for classification and recognition needs to be further explored and compared with the existing methods. Of course, because the CNN algorithm has some requirements for the amount of data, as your article said, these still need to enrich the database and further research.

Response: We wholeheartedly agree with the Reviewer on the three points, all of which need further investigations.  (1) Regarding the deviations between simulated and actual data, model validation with in vivo images will be most relevant.  Before clinical trials, more realistic modeling will be pursued, including image-based lung models with more branching generations and disease-induced airway remodeling.  (2) Regarding the feature loss during feature extraction, the original images should be used in place of, or in addition to, the extracted features to develop classifiers. (3) The requirement of a sufficiently large database for meaningful CNN will no doubt be a hurdle to this method.  The bright side is that this hurdle will be gradually eased as the database grows.         

Reviewer 3 Report

It's an interesting study covering the two main aims of JNT - Point-of-care systems for personalized health care and Personalized nanomedicine. The manuscript is well structured and is the output of the many years of research of the authors on the presented topic. I recommend acceptance of the manuscript for publication in JNT.

Author Response

It's an interesting study covering the two main aims of JNT - Point-of-care systems for personalized health care and Personalized nanomedicine. The manuscript is well structured and is the output of the many years of research of the authors on the presented topic. I recommend acceptance of the manuscript for publication in JNT.

Response: We deeply appreciate the Reviewer’s support for this study.